# The carbon footprint of the U.S. multinationals' foreign affiliates

Luis-Antonio López[1], María-Ángeles Cadarso[1], Jorge Zafrilla [1] & Guadalupe Arce[2]

Multinational enterprises (MNE) need to be a part of the solution in the fight against climate change, as claimed by investors and consumers, reducing emissions within their operations and supply chains. This paper measures the carbon footprint of U.S. MNE foreign affiliates (US-MNE) operating beyond the U.S. borders. Using a multiregional input-output model and information about US-MNE activities, the US-MNE carbon footprint ranks US-MNE as the 12th top emitter of the world. In relative terms, one dollar of value added generated by US-MNE affiliates operating abroad requires higher emissions than the domestic average and the ratio increases when only developing host countries are considered. Only 8% of total carbon footprint returns to the U.S. as virtual carbon embodied in the U.S. final consumption. Potential technology transfers between the U.S. parent company and affiliates to reduce US-MNE carbon footprint have been performed to evaluate potential rippled effects of mitigation actions.

[1] Faculty of Economics and Business, University of Castilla-La Mancha, Plaza de la Universidad, 1, 02071 Albacete, Spain. [2] Faculty of Economics and Business, Complutense University of Madrid, Campus de Somosaguas, 28223 Pozuelo de Alarcón, Madrid, Spain. Correspondence and requests for materials should be addressed to L.-A.L. (email: Luis.LSantiago@uclm.es) or to M.-Á.C. (email: Angeles.Cadarso@uclm.es) or to J.Z. (email: Jorge.Zafrilla@uclm.es) or to G.A. (email: Garce01@ucm.es)

The Paris national pledges (the so-called Nationally Determined Contributions, NDCs) are not sufficient to meet temperature targets[1,2]. There is an urgent need to involve more participating agents[3] to create integrated systems seeking global sustainability[4,5]. Different countries' courts have started to force governments to obey their own air-quality laws and firms to meet their legal obligations to manage climate risks[6]. In a sense, all public and state agents have to establish strong commitments in the fight against climate change to overcome setbacks such as the recent Paris withdrawal led by President Trump[7]. The states of the U.S. must use previous experience to continue to be the primary drivers in reducing power-sector $CO_2$ emissions in the U.S.[8]. The mayors of different cities of the world have taken a step forward by committing to the Global Covenant of Mayors for Climate and Energy. To date, only 157 U.S. mayors are involved[9].

In the private economic sphere, investors are requiring better quantitative assessments of how firms/multinational corporations are managing climate risks and reporting information regarding mitigation and adaptation to climate change[10]. Researchers are proposing principles to help investors select asset portfolios consistent with long-term climate goals[11]. The Task Force on Climate-related Financial Disclosures (TCFD) released three key documents by June of 2017 to measure and respond to climate change risks and to encourage firms to align their disclosures with investors' needs[12]. In February of 2018, over 390 organizations and the CEO climate leaders at the World Economic Forum supported the recommendations of the TCFD. The number of corporations committed to climate actions has grown steadily over the last few years, as recorded by various networks and platforms[13]. Only NAZCA platform recorded actions from 2.431 companies in 2018[14] such as emissions reduction pledges, commitments to renewable energy production and to setting internal carbon prices[15,16].

Among firms, multinational enterprises (MNE), by their economic power, the fact that their control is highly concentrated in the hands of a few top holders[17] and their transnational character, must be key agents in the fight against climate change. Nearly two-thirds of historical $CO_2$ and methane emissions during the period 1854–2010 can be attributed to 90 incorporated entities that produce energy and cement[18]. However, none of the multinational firms accompanying former President Obama at the 2015 COP21 Conference belonged to energy intensive sectors[19]. This is not a paradox because, indirectly, the decisions taken by other thousands of firms have determinant effects on global warming. In other words, a significant part of the environmental impacts of many firms are generated beyond their borders along their global value chains (GVCs) but are driven and enhanced by those firms' decisions. Those decisions represent opportunities for emission reductions, such as the following: creating sectoral standards that use incentives or sanctions to help operationalize codes of conduct across the global supply chains[20–22]; choosing suppliers intensive in low carbon technologies[23], because of the global environmental benefits these technologies present[2]; choosing more environmentally friendly distributors (downstream) with less income-based carbon emissions[24]; transferring of technology to their suppliers/partners in other countries[25]; or designing products and improving existing ones to minimize material and energy use[26], facilitating a circular economy[27]. However, only 13% of recorded mitigation actions of firms address the whole supply chain, including scope 3[15].

The significance of the environmental impacts of firms, particularly in the case of MNE, not only transcends the limits of the firms themselves but also stretches beyond the borders of their country of origin because of the effect of the international trade, foreign direct investments and income and production generated elsewhere. The importance of international trade and global value chains in terms of carbon emissions has been profusely researched[28–31]. The trade relationships between developed and developing countries have led to an unequal ecological exchange that threatens mitigation commitments[32], with a recent growth in carbon embodied in South-South trade[33] and eases the existence of the pollution haven hypothesis[34]. In the case of the U.S. economy, a reduction in 2007–2013 domestic $CO_2$ emissions has been mainly motivated by the economic recession and changes of other factors, such as international trade and fuel mix, have played a minor role[35]. The worldwide redistribution of $CO_2$ emissions and other pollutants has also had harmful effects on Chinese health[36] and on U.S. West Coast citizens because of the atmospheric transportation of pollutants from emission-intensive regions such as China[37].

The main contribution of this research is the calculation for the first time of the carbon footprint of US-MNE operating elsewhere through the generation of value added by its affiliates. Differentiating total carbon responsibility of firms according to ownership and size can be relevant in the policy context, but most of this kind of research focus on some aspects of the Chinese economy only[38–40], so far.

The impact in terms of carbon emissions of MNE can no longer be ignored, as majority-owned U.S. foreign affiliates generated a value added of $894.5 billion and employed 6.8 million workers during 2015, which represents, respectively, 6.4% and 5.5% of U.S. private industry[41]. The foreign US-MNEs' carbon footprint calculation proposed in this paper quantified 0.5082 $GtCO_2$ during 2009, the virtual carbon embodied in the production of final goods and services supplied by MNE corporations producing and generating added value and employment abroad. The economic activity of firms such as Amazon, Google, and McDonalds are behind these estimations. This US-MNE carbon footprint represented 1.5% of the worlds' global emissions during 2009. If foreign US-MNE activity were to be cataloged as a country, it would be ranked as the 12th top emitter in the world. This measure allows us information regarding the US-MNE addiction to carbon[42], the environmental sustainability/viability of their business and the risk to stakeholders if these firms do not meet society's carbon expectations.

## Results

**US-MNE affiliates' carbon footprint.** The activity of US-MNE beyond U.S. borders accounts for 0.5082 $GtCO_2$, which represents 9.8% of U.S. producer responsibility (PR) and 8.6% of the U.S. consumer responsibility (CR), and 1.5% of the world's global emissions during 2009 (Fig. 1). The amount of the foreign affiliates US-MNE carbon footprint was estimated using the value added generated by the US-MNE abroad as an indicator of the generation of burden shifted income and emissions (see Supplementary Table 1).

One dollar of value added generated by US-MNE affiliates operating abroad requires higher emissions (0.46 $kgCO_2$/$) than that at home (0.37 $kgCO_2$/$). As we used the host sector and country production technique for the MNE, the results indicate that US-MNE are polluting more in countries and sectors than the average at home. In addition, we observed a relative territorial mismatch, or international imbalance (unequal exchange)[43], as long as the U.S. private industry value added generated abroad by the US-MNE is ~7.8%, while the total amount of emissions generated by that activity in relation to the U.S. PR is 1.3 times higher. As previously remarked, the foreign affiliates US-MNE carbon footprint per unit of value added generated reached 0.46 $kgCO_2$/$ but varied depending on the country where it was produced. In the case of emerging economies and Canada, the ratio grows to 0.62 $kgCO_2$/$ and 0.48 $kgCO_2$/$, respectively.

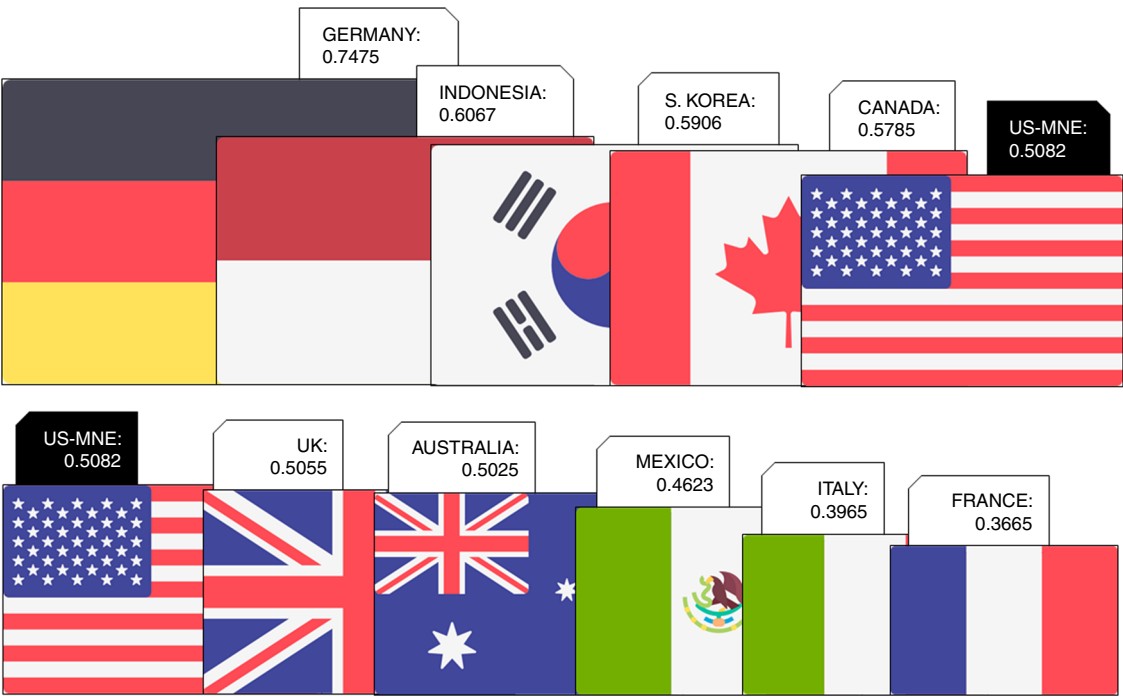

**Fig. 1** 2009 U.S. Multinationals carbon footprint among the world ranking of emitters (GtCO$_2$). See Supplementary Data 1. The flags icons in Fig. 1 are made by Freepik from www.flaticon.com

However, in the case of Europe, we observed a lower ratio, 0.36 kgCO$_2$/\$. As a result, with the exception of Europe, the remaining cases show the existence of the pollution haven hypothesis (PHH)[34], as the economic average U.S. PR and CR per unit of value added reached 0.37 and 0.41 kgCO$_2$/\$, respectively, lower than the foreign MNE values.

For perspective, if foreign US-MNE activity was to be cataloged as a country, it would be ranked the 12th top emitter in the world (Fig. 1, and Supplementary Table 1). The amount of emissions generated by the foreign activity of US-MNE is similar to that of United Kingdom or Australia (+1% higher) and greater than the total PR of countries such as Mexico (+9%), Italy (+22%), France (+28%), Poland (+32%), Turkey (+40%), Taiwan (+40) and Spain (+44%) during 2009. In this list, the US-MNEs' footprint is only below to the eleventh largest polluting countries, closer to countries such as Canada (−14%) and Korea (−16%). Although foreign US-MNEs' carbon footprint is far from that of the top emitting countries, the intermediate position that they achieved placed polluting US-MNE activity abroad among some of the richest and most developed economies in the world.

**US-MNE affiliates' producer footprint**. Under the host country principle, the US-MNEs' footprint is allocated to the country where the multinational is located, defined as producer footprint (PF) in the methods section. The US-MNEs' footprint is concentrated in the European countries as a whole, accounting for more than 35.7% of the total US-MNE footprint (Fig. 2, and Supplementary Table 2). Separately, countries such as the United Kingdom, Germany, Ireland and France account for 7.7, 7.2, 4.3, and 4.1%, respectively, the third, fourth, seventh, and eighth ranked, respectively, of the total US-MNE footprint. The top of the list region is ROW accounting for more than 25% of the total US-MNE footprint, down the list, the second region is Canada with almost 11% of the footprint. China is in fifth place, accounting for 7% of the total US-MNE footprint. Leaving aside the case of the European countries and the strong economic

interrelationship between them and the U.S., the emissions generated by US-MNE among emerging economies accounts for 48% of the total emissions. This result deepens the geographical/international mismatch already seen with the global figures, as the value added generated in these emerging economies accounts for only 35% of the total value added generated by US-MNE abroad. On the one hand, greater emissions in emerging/developing economies result from greater emission intensities than those of developed countries. On the other hand, in the case of European developed economies, an international imbalance is also observed, but it moves in the other direction. While nearly 46% of value added is generated in Europe, only 36% of the emissions are generated in European regions.

**Weighting US-MNE affiliates' producer footprint**. When the foreign host country US-MNEs' footprint is weighted by each country pollution intensity, small countries such as Ireland and Luxembourg move much higher to the top positions of the list. Ireland serves as a paradigmatic case (see the complete list of countries in the Supplementary Table 3). More than 48% of the Irish PR is generated by the activity of US-MNE measured by the value added generated (Fig. 3). In Luxembourg, US-MNE activity accounts for >31%. The remaining countries account for <10% of each of their PR generated because of the activity of US-MNE; the order of the list descends as follows: Canada (9.6%), the United Kingdom (7.7%), and Belgium (7.7%). The emergence of two small countries in terms of US-MNE footprint, which can be considered as tax havens, is relevant in our analysis, as it represents a large amount of emissions in each county. For the remaining top ten positions, with the only exception being Canada (third) and Australia (tenth), we found EU countries ranging from the United Kingdom in fourth place to Germany in ninth place (4.9%). The first emerging economies on the list are Mexico, with 3.1% of Mexico's PR, followed by Brazil with 3.0% and Turkey with 2%. Throughout the value added generated by US-MNE abroad, following a replica localization strategy, as in

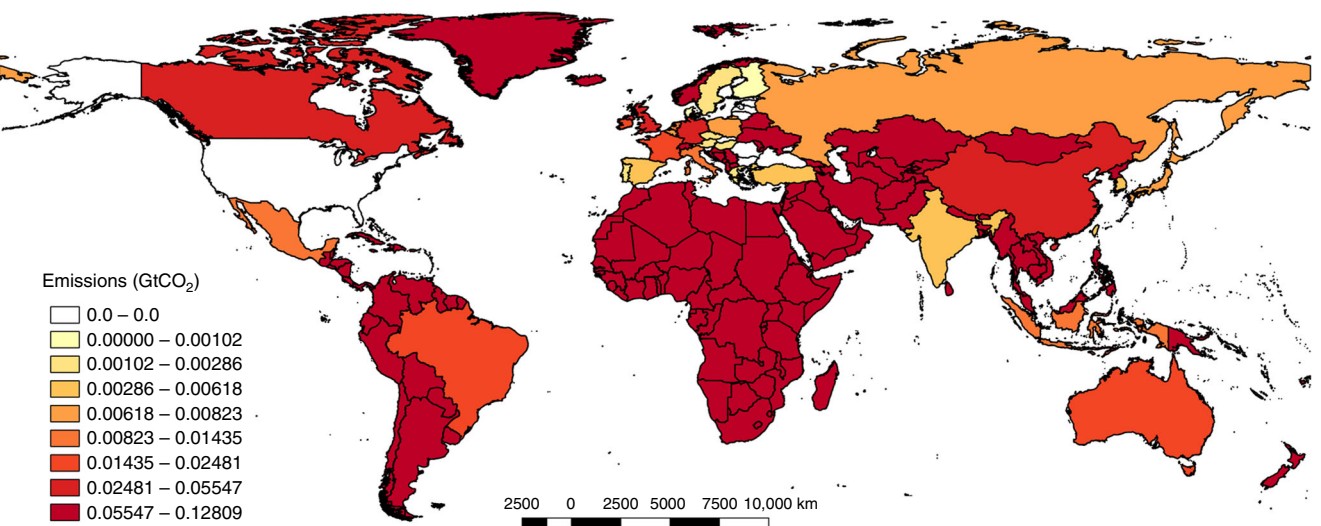

**Fig. 2** US-MNE carbon footprint by host country, the producer footprint (GtCO$_2$). This study covers the following 27 EU countries and 13 other major countries (using ISO 3166–1 alfa-3 country codes): AUS, AUT, BEL, BGR, BRA, CAN, CHN, CYP, CZE, DEU, DNK, ESP, EST, FIN, FRA, GBR, GRC, HUN, IDN, IND, IRL, ITA, JPN, KOR, LTU, LUX, LVA, MEX, MLT, NLD, POL, PRT, ROU, RUS, SVK, SVN, SWE, TUR, TWN, and USA. The remaining non-covered part of the world is called ROW. See Supplementary Data 2. This map has been created using QGIS Geographic Information System. Open Source Geospatial Foundation Project. https://qgis.org

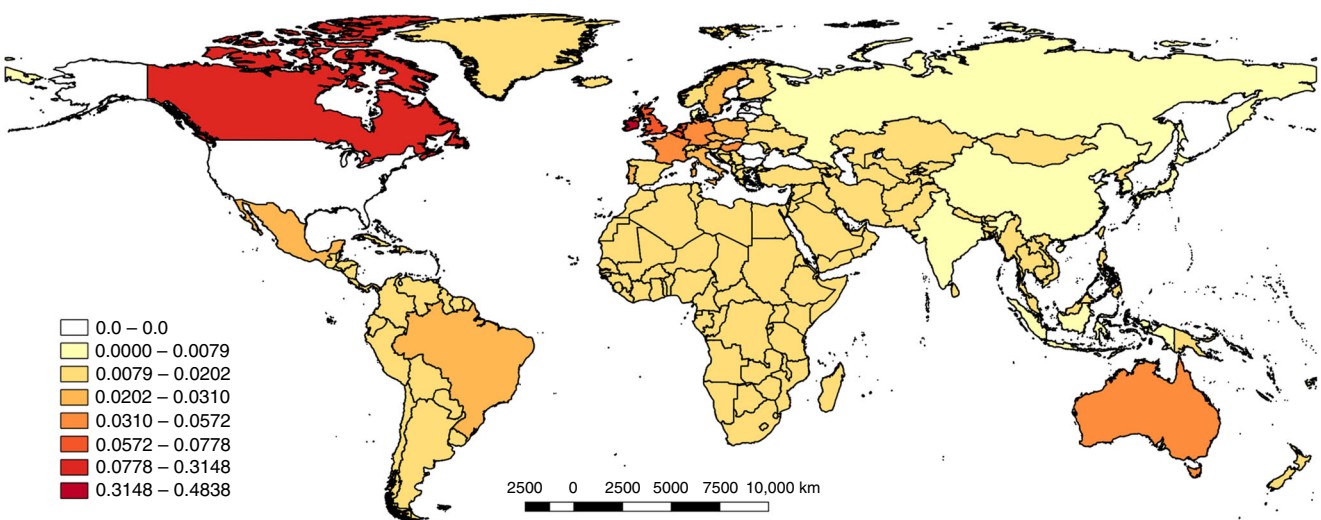

**Fig. 3** 2009 weight of US-MNEs' carbon footprint on producer responsibility of the host country (producer footprint). This study covers the following 27 EU countries and 13 other major countries (using ISO 3166–1 alfa-3 country codes): AUS, AUT, BEL, BGR, BRA, CAN, CHN, CYP, CZE, DEU, DNK, ESP, EST, FIN, FRA, GBR, GRC, HUN, IDN, IND, IRL, ITA, JPN, KOR, LTU, LUX, LVA, MEX, MLT, NLD, POL, PRT, ROU, RUS, SVK, SVN, SWE, TUR, TWN, and USA. The remaining non-covered part of the world is called ROW. See Supplementary Data 3. This map has been created using QGIS Geographic Information System. Open Source Geospatial Foundation Project. https://qgis.org

the case of McDonald's and Starbucks, or selling goods and services, as in the case of Amazon and Apple, the MNE internationally transfer their environmental responsibility mainly to developed countries with greater markets or greater purchasing power instead of transferring them to developing countries.

**US-MNE affiliates' responsibility, a consumer perspective**. Alternatively to the host country perspective (PF), the US-MNEs' footprint can be allocated not only to the host country but also to the consumer country, where goods are finally consumed, according to the consumer-based approach (CF). Of the emissions linked to the US-MNEs' sales of goods and services, 61% occur within the borders of the country where they are produced,

and 39% are exported as final products during the last production round. Higher consumer US-MNE footprints are those of the ROW (19% of the total footprint and 96,378 KtCO$_2$), Canada (9.4%) and Germany (7.6%). However, the most notable appearance on the list of countries (third) with a higher consumer US-MNE carbon footprint is the U.S. (7.8% of total CR), explained by the import of 39,616 KtCO$_2$ embodied in goods and services produced by its MNE abroad that returned to be ultimately consumed in the U.S. (Fig. 4, and Supplementary Table 4).

Figure 5 shows the top 200 carbon flows, representing 78% of the total flows related to US-MNEs' exports operating abroad. The stream size shows the importance of the carbon flows embodied in exports, and the arrow shows the origin of the country where the multinational is operating. The most relevant

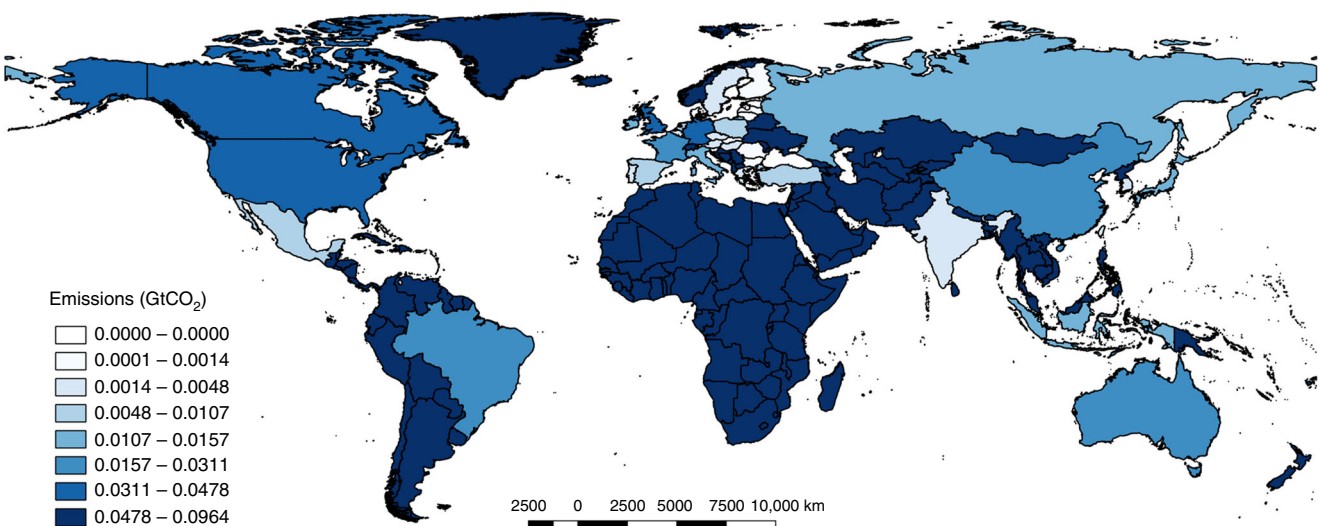

**Fig. 4** 2009 US-MNEs' carbon footprint by Consumer Country (MNE CF) (GtCO$_2$). This study covers the following 27 EU countries and 13 other major countries (using ISO 3166–1 alfa-3 country codes): AUS, AUT, BEL, BGR, BRA, CAN, CHN, CYP, CZE, DEU, DNK, ESP, EST, FIN, FRA, GBR, GRC, HUN, IDN, IND, IRL, ITA, JPN, KOR, LTU, LUX, LVA, MEX, MLT, NLD, POL, PRT, ROU, RUS, SVK, SVN, SWE, TUR, TWN, and USA. The remaining non-covered part of the world is called ROW. See Supplementary Data 4. This map has been created using QGIS Geographic Information System. Open Source Geospatial Foundation Project. https://qgis.org

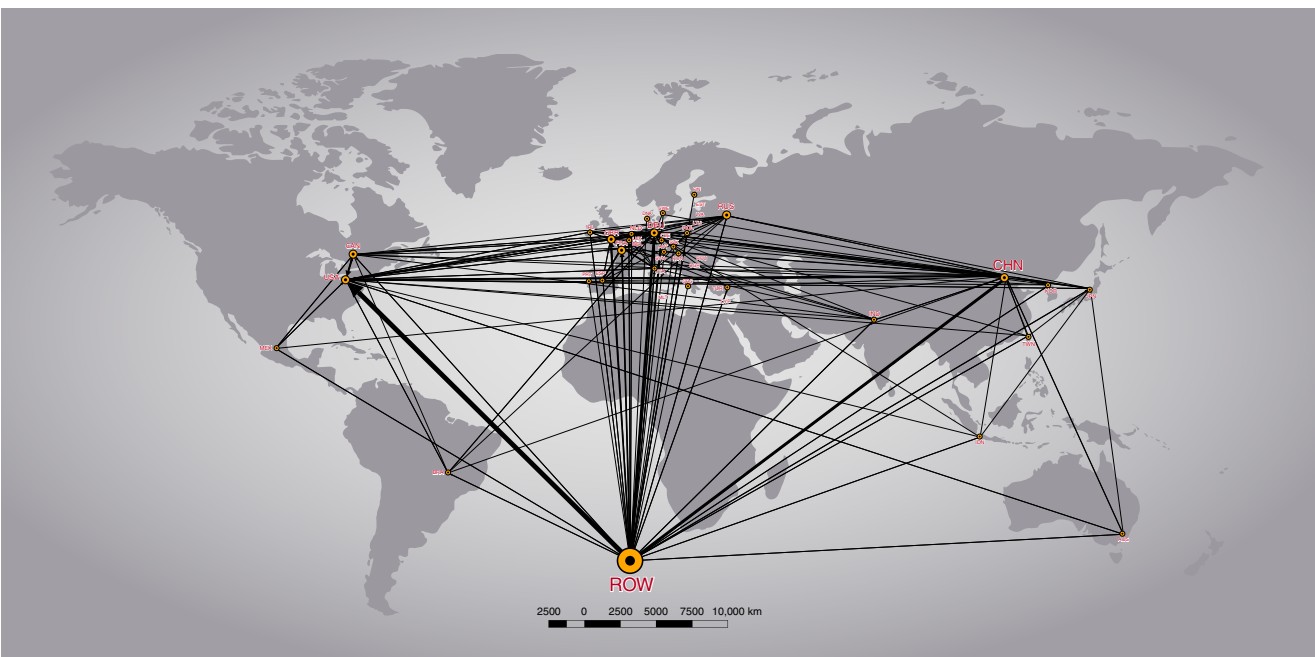

**Fig. 5** Carbon flows embodied in US-MNEs' exports operating a broad. This study covers the following 27 EU countries and 13 other major countries (using ISO 3166–1 alfa-3 country codes): AUS, AUT, BEL, BGR, BRA, CAN, CHN, CYP, CZE, DEU, DNK, ESP, EST, FIN, FRA, GBR, GRC, HUN, IDN, IND, IRL, ITA, JPN, KOR, LTU, LUX, LVA, MEX, MLT, NLD, POL, PRT, ROU, RUS, SVK, SVN, SWE, TUR, TWN, and USA. The remaining non-covered part of the world is called ROW. Regions and countries with greater nodes are greater receivers of emissions flows embodied in international trade (US and ROW in red, followed by Germany in orange, and by the United Kingdom, France and India in brown). The ROW is arbitrarily placed at the bottom of the map for the sake of presentation. See Supplementary Data 5. This Figure has been created using the software Gephi, developed in Bastian et al.[71]

flows are those from the ROW to the U.S. and Germany, from China to the ROW and the U.S. and from Canada to the U.S. The difference between the host and consumer country US-MNE carbon footprint shows the foreign US-MNEs' carbon balance by country, meaning the international emissions balance of carbon flows embodied in goods and services internationally traded that are produced by US-MNE and sold beyond the producer country's borders as final exports minus the final imports

(Supplementary Table 4). The most relevant carbon net exporters are the ROW region (showing a positive emission balance of 31,716 KtCO$_2$), followed by Canada, Ireland, The Netherlands and China. Notably, these countries are used by US-MNE more as platforms to produce for other markets (because of their natural resources; lower costs, including taxes or environmental regulation; or proximity to bigger markets), and because of the presence of US-MNE, they are polluting for others' consumption.

However, after the U.S., the main carbon net importers are Russia (with a negative emissions balance of 5,047 KtCO$_2$), Japan, Italy, Spain, and France. In these countries, the emissions embodied in imports from US-MNE are higher than those embodied in the country's US-MNE production at home (with the exception of the U.S. where only imports are considered by definition). These countries that have negative balances regarding US-MNE are benefiting from other host countries' MNE polluting activities, increasing at the same time their carbon footprint.

If we weigh the consumer US-MNEs' carbon footprint according to each country's CR, most of the top 20 positions of the list are occupied by the largest European countries in terms of GDP, with the exception of third place Canada (7.99%) (see Supplementary Table 5). The case of Ireland is still paradigmatic (similar to the case of PR measures), as the US-MNEs' carbon footprint represents 26.4% of Ireland's CR, followed by Luxemburg at 10.5%. Largest emerging economies appear even further down the list, as in the cases of the Brazil in the 13th position or China and Mexico at 41st and 20th, respectively. The U.S. also appears lower on the list, as its foreign US-MNEs' carbon footprint accounts for only 0.67% of its CR. The relevance of imports from US MNE in affluent European markets along with the lower average carbon intensity of European products explain this result.

**US-MNE affiliates' footprint by sectors**. The US-MNEs' footprint is concentrated in industrial sectors (Fig. 6) both in developed (90%) and developing/emerging countries (91%): other industries (24%), remainder of manufacturing (22%), chemicals (14%), food (9%), and transportation equipment (7%). This concentration can be explained by the high direct carbon intensity of several industrial sectors (chemical products and electricity and refined petroleum products are included in other industries) and by the high consumption of intensive energy inputs (transport equipment and the remainder of manufacturing), similar to the distribution pattern of virtual carbon embodied in international trade[30]. Among service activities, we could highlight the US-MNEs' footprint in distribution sectors such as Wholesale trade and Retail trade, although it reaches only 4% of the total US-MNE carbon footprint.

The generation of a large amount of value added abroad by US-MNE does not imply a generation of great carbon footprints because of the differences in pollution intensities among activities. In this sense, although wholesale trade accounts for 13% of the value added, it generates only 4% of the US-MNE footprint. The

carbon intensity per unit of value added generated by this sector is quite low, 0.12 kgCO$_2$/\$ in developed countries and 0.19 kgCO$_2$/\$ in developing/emerging economies (see Supplementary Tables 6 and 7). However, food and other industries represent only 3 and 7% of the total value added and 9 and 24%, respectively, of the US-MNEs' footprint. The emissions multipliers are higher in these sectors: 0.99 kgCO$_2$/\$ and 1.30 kgCO$_2$/\$ in developed countries and 2.29 kgCO$_2$/\$ and 3 kgCO$_2$/\$ in developing/emerging countries. The MNE activity in developed countries presents lower pollution intensities, per unit of value added, than in emerging economies in most sectors. The only exception is the mining industry, with a higher carbon intensity in developed countries (0.08 kgCO$_2$/\$) than in emerging economies (0.03 kgCO$_2$/\$) as a consequence of the greater natural resources endowment in emerging economies, which allows for less pollution intensive extraction processes[34]. Hence, although mining is the sector with the highest figure in terms of value added generated by US-MNE in emerging economies, reaching 106,266 M\$, the carbon footprint generated reaches only 3680 ktCO$_2$, 2% of the total footprint in emerging countries.

**Scenarios by indicator and policy recommendations**. The estimation of the US-MNE carbon footprint is based on two main assumptions: the selection of the value added they generate in the host country as an indicator of MNE activity and the consideration that the production and pollution structure of the MNE is the same as that of the hosting sector and country. This section explores the scope of these assumptions by considering other alternatives for both and evaluating potential policy implications of a technology transfer strategy. One limitation of the value-added data is the possibility of measurement error because of transfer pricing. The BEA does not have any established methodologies to account for this[44]. To account for this limitation, in addition to value added, the estimation of the US-MNEs' footprint can be extended to other indicators to capture, in an alternative way, the foreign activity of the MNE. Foreign US-MNE activity could be approximated by indicators; in monetary terms, such as compensation of employees and capital compensation; and in physical terms, employment. All of these indicators refer to the activity of US-MNE within the borders of the host countries[45], producer footprint. Regarding production (and pollution) techniques, our baseline scenario ties host country production and pollution structure to US-MNE, assuming no technology transfer. This assumption is not as robust in the case of emerging economies given their greater carbon intensities[34], as

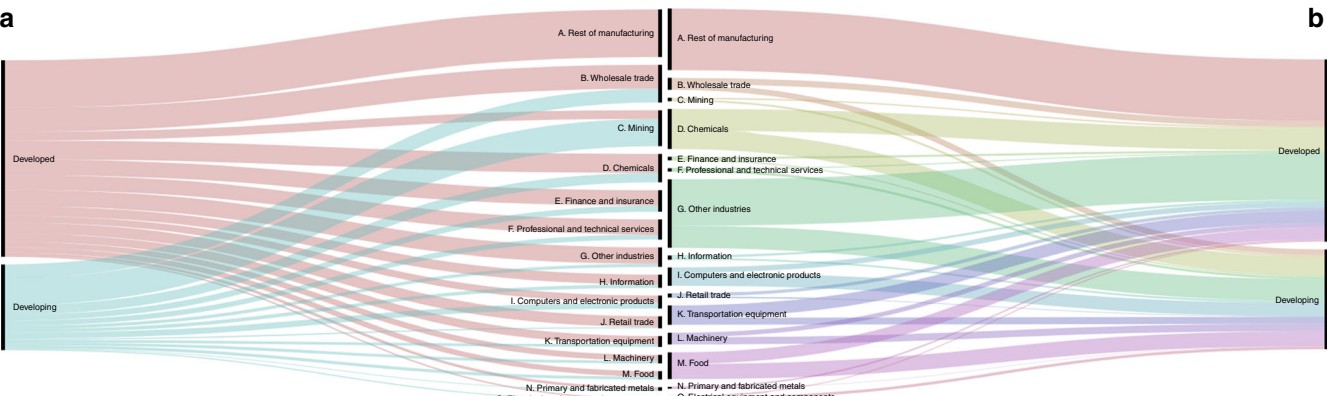

**Fig. 6** Shares of US-MNEs' value added and producer footprint by sectors (%). Value added generated (**a**) by US-MNE abroad is 709,686 million \$ in developed economies (65%) and 389,634 million \$ in developing economies (35%). The carbon footprint (panel b) of US-MNE affiliates is 265,175 Kt CO$_2$ (52%) in developed countries and 243,039 Kt CO$_2$ in developing countries (48%). See Supplementary Data 6. This Figure has been created using the open source data visualization framework RAW Graphs, developed in Mauri et al.[72]

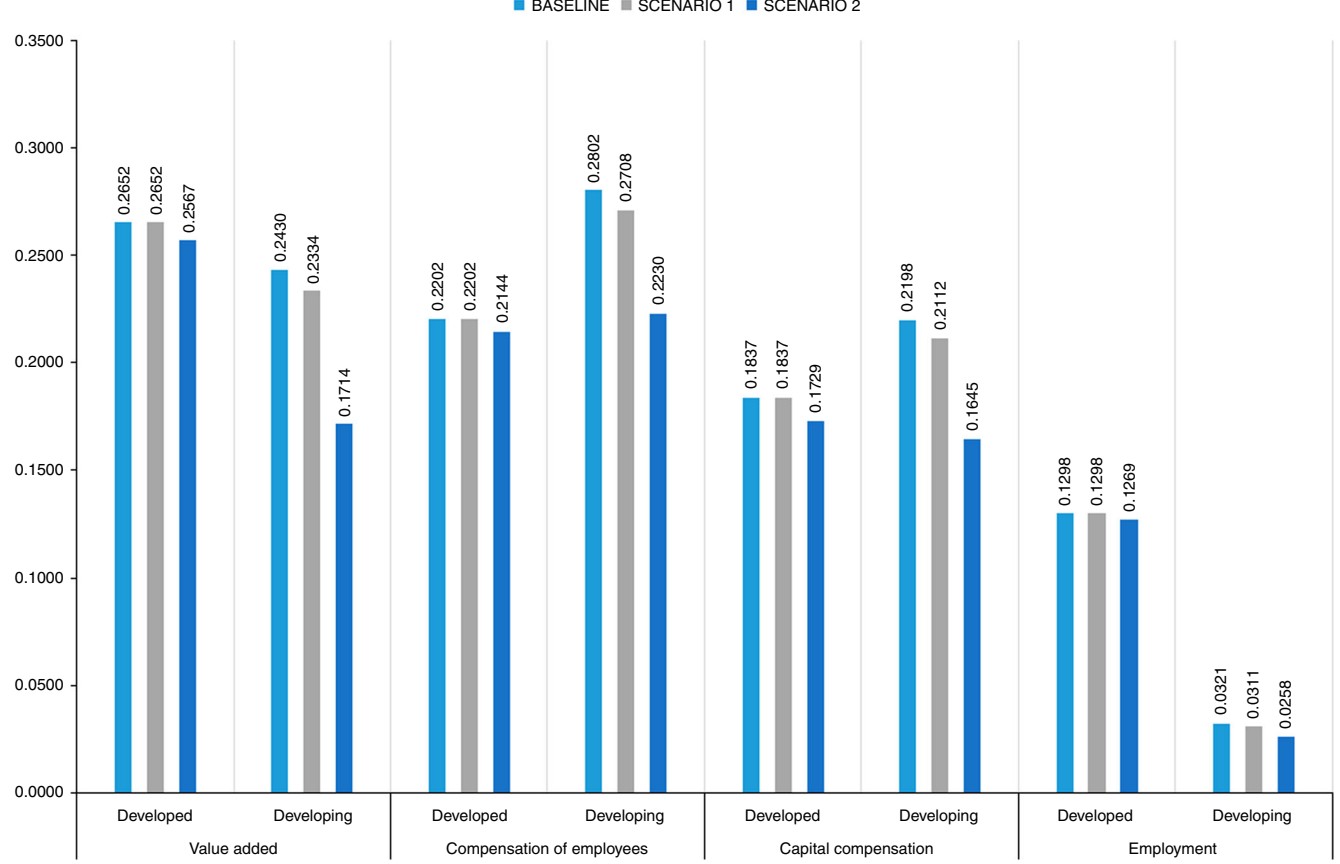

**Fig. 7** 2009 US-MNEs' carbon footprint in developed and developing countries by indicator and scenario (GtCO$_2$) See Supplementary Data 7

they act only as export processers in many cases[38–40]. In this sense, we extend the baseline approach by proposing two additional scenarios: (1) replacing the production structure of the emerging countries with US technology of production and (2) the same scenario but one that also replaces the emerging countries' carbon emission intensities with U.S. pollution intensities.

Focused on the baseline scenario and dividing the US-MNEs' activities between developed and developing host countries, the estimation by monetary indicator of the US-MNEs' footprint in developed countries ranges between 0.2652 GtCO$_2$, using the value added, to 0.1837 GtCO$_2$, using capital compensation ranging from 5.1% to 3.5% of the total U.S. PR (Fig. 7). In developing countries, the US-MNEs' footprint ranges from 0.2802 GtCO$_2$, using compensation of employees, to 0.2198 GtCO$_2$, using capital compensation ranging from 5.4% to 4.2% of the total U.S. PR. The variability of US-MNE footprint over all countries, using value added, ranges from 9.8% to 7.8% of the total U.S. PR. The footprint estimation using the capital compensation indicators generated by US-MNE in developed countries is nearer the values shown by the compensation of employees' indicators than those in developing countries, showing that a greater amount of US-MNEs' capital compensation over compensation of employees is generated in developed countries.

In physical terms, using employment (thousands of employees) as an indicator, the results range from 2.5% in developed countries to 0.6% in developing countries of the total U.S. PR, which shows a type of incompatibility with the monetary indicators because of the lower results. The lower US-MNE carbon footprint based on the employment indicator, both for developed and developing countries, shows greater US-MNE labor productivity than that in the host countries' firms. The

lower US-MNE footprint associated with the employment indicator, compared to that associated with the compensation of employees at the US-MNE, shows in average terms that although US-MNE are not intensive in terms of employment generation beyond their borders, they pay higher wages than domestic firms.

In this context, the implementation of a technology transfer policy between the U.S. parent company and affiliates could lead a reduction of the carbon footprint. If the production of the US-MNE foreign affiliates took place within the U.S. borders, the amount of emissions would be almost the same (counterfactual 0.5017 GtCO$_2$, compared to US-MNE carbon footprint 0.5082 GtCO$_2$). However, scenarios 1 and 2, simulate changes in the production and pollution structures of only developing countries from those of the host country and sector to that of the U.S. parent counterpart, and result in lower figures in terms of footprint for all indicators (Fig. 7). In terms of footprint, the changes in emissions coefficients in emerging economies slightly improve in the developed countries' footprint by a reduction of 2.2% using employment as an indicator and by 5.9% using capital compensation. The case of developing countries is different; the results of scenario 1 show lower footprints for all the indicators, from −4.0% in the case of value added to −3.2% in the case of employment. Greater reductions are observed in scenario 2 when carbon emissions intensities are replaced. In this case, the total reduction observed by the indicator is as follows: value added −29.5%, employment −19.5%, compensation of employees −20.4%, and capital compensation −25.1%. The measures provided by scenarios 1 and 2 compared to the baseline provide an estimation of the mitigation potential of US-MNE throughout the world through home technology deployment. Although

cleaner technology transfers between headquarters and the emerging countries could lead to reductions in the US-MNEs' footprint, the wide differential observed between scenarios 1 and 2 shows that the most relevant transfer has to be led by changes in the use of intensive fossil energy resources. In this sense, the US-MNE could draft contracts forcing suppliers to certify the use of cleaner energy sources prior to allocation.

## Discussion

Natural boundaries that limit climate change are set by our planet. However, human impacts on climate change are associated with the economic activity (business production, household consumption, investment financing, public spending, etc.) that occurs within a political jurisdiction (state, region or city). The proposed procedure to calculate MNEs' carbon footprint reaches beyond these limits and assesses MNEs' environmental responsibility through global value chains regardless of where their activity occurs or where their goods are consumed. The aim was to present the impact of multinationals in a manner that provides sufficient information for sustainable management, investment, and consumption.

The MNEs' carbon footprint shifts the focus from the role of countries as producers or consumers[28,29,46] to the role of multinational as transnational entities, specifically, to the value added generated by MNE producing beyond U.S. borders. Currently, some alternative emissions allocation criteria are broadening literature beyond the well-established producer-based and consumption-based accounting systems. Those criteria share emissions responsibilities among the economic agents. Some examples are the income-based accounting[47], the technology adjusted consumption-based approach[48,49]. Our results provide information that could be used at the country level as an allocation key to share emissions responsibilities, allocating the outward responsibility of the MNE to the headquarters' country. This could be a first step for the estimation of a shared responsibility criterion based on the control performed by MNE[50]. In addition, the explicit consideration of final direct investment of MNE abroad would be an additional element to be evaluated in terms of carbon emissions by the capital endogenization in a context of responsibility allocation criteria[51] or by integration of capital investment and depreciation in a dynamic carbon footprint model[52].

Through the MNE carbon footprint, the firm becomes knowledgeable regarding the environmental impacts of its whole production network and thus could take responsibility for the external costs associated with these impacts around the world (both where the firm is producing, the host country, and where its products are being consumed); this would help in the implementation of greener global supply chain management[4]. The existence of a small number of important carbon clusters, which are part of the global supply chains network related to the worldwide economy's final demand, could facilitate carbon control[53] and would allow for the inclusion of emissions along the global value chain (that is, scope 3). Firms' reports on corporate social responsibility are only beginning to estimate scope 3 impacts from 2013[54], and their monitoring differs depending on each corporate sector[55].

Although countries are not allowed to legislate beyond their borders, corporations can transfer technical specifications to their foreign suppliers, avoiding border legislation restrictions or searching for sector coordination[21,22]. The scope of influence of the firm is not only their own production process but also those of their outwards suppliers. The measure of MNEs' carbon footprint provides a total potential mitigation effect that could ripple through the economy (of the host or consumer country). A

limitation of our results is that we do not consider the influence of all U.S. firms on emissions embodied in international trade, as we are only considering MNE generating value added abroad. The rest of U.S. firms' outsourcing that only imply importing goods and services without generating own value added in the producer country are not included. For the Chinese case, we would only be accounting up to 54% of emissions embodied in exports in 2010 induced by foreign-owned firms of different contries[56], since the remaining 46% is induced by companies with Chinese capital. In terms of mitigation policies, focusing on MNE, that wiled the production control compare to firms that only outsource or import intermediate goods, have several advantages: (a) transaction costs between parent and affiliate companies are lower compared to firms with different owners[57], which facilitates the transmission of information and lower carbon intensive production standards; (b) outsourcing or importing companies, in as much as represent a small share of the sales of suppliers, have lower bargaining power compared to those affiliate companies that supply their production to other group companies.

Once multinationals know their carbon footprint, not only can more sustainable management be addressed, but MNE can also transmit that information to stakeholders on the one hand and to consumers on the other hand with the objective of reducing their reputational and investment risk and improving the loyalty and choices of consumers, who are able to guide the global economy onto the path of sustainability across their consumption decisions[58].

## Methods

**Footprints and the new concept of producer footprint**. Currently, the multiregional input-output method to calculate the carbon emissions responsibility of a country is well established both from the producer perspective (PR) and from the perspective of the consumer (CR)[29,59]. We used the basic input-output expression and extended it to consider carbon emissions, substituting the emissions coefficient vector and final demand with their diagonal counterparts ($\hat{\mathbf{e}}$, and $\hat{\mathbf{Y}}$, respectively). Depending on how the final demand was diagonalized, we obtained a different allocation for the carbon footprint. We considered two possibilities.

First, if the final demand is diagonalized by sections[32], meaning that the domestic final demand is shown in the main diagonal, while the final exports/imports are shown in off-diagonal positions ($\hat{\mathbf{Y}}$), we obtain a matrix of total emissions ($\mathbf{F}$) $\left(\mathbf{F} = \hat{\mathbf{e}}(\mathbf{I} - \mathbf{A})^{-1}\hat{\mathbf{Y}}\right)$. Summing the $\mathbf{F}$ matrix by rows results in total emissions (domestic) per production country, PR, and is the measure considered by the Kyoto Protocol and the Paris Agreement for country commitments to emissions reduction. Summing along the columns, we have emissions generated all over the world linked to one country's final demand, CR.

Second, if final demand is just the diagonalized row sum of the Y matrix $\left(\mathbf{y} = \sum_s \mathbf{Y}^{rs}\right)$, placed in the main diagonal, we obtain a matrix of total emissions, $\bar{\mathbf{F}}$ $\left(\bar{\mathbf{F}} = \hat{\mathbf{e}}(\mathbf{I} - \mathbf{A})^{-1}\hat{\mathbf{y}}\right)$. Matrix $\bar{\mathbf{F}}$ shows the same amount of emissions by country (and sector) as the F matrix and the same PR, when summing along the rows. However, the results by columns are different because $\bar{\mathbf{F}}$ implies the allocation of emissions embodied in exports to the country of production instead of to the country of consumption as in $\mathbf{F}$. Summing along the $\bar{\mathbf{F}}$ columns, we quantify the carbon emissions, domestic and imported, embodied in the production of goods and services that the industry provides to the final demand (including exports). We call this measure the producer carbon footprint, $\mathbf{PF}$ $\left(\mathbf{PF} = \mathbf{f}^s \sum_r \bar{\mathbf{F}}^{rs}\right)$, because it is based on the producer's (or industry's) global responsibility on emissions and it focuses largely on the industry and its needs for inputs (direct and indirect, including those imported) for its whole production (including exports). As a result, the $\mathbf{PF}$ measure goes a step further in the need of firms to achieve sustainable management throughout their global production chains and then transfers their achievements to customers and final consumers throughout the world.

**MNEs' carbon footprint by host and consuming country**. The procedure we proposed for the calculation of one country c's MNE (which is property of the country c) carbon footprint is conditional on the lack of information regarding technology and trade flows of intermediate and final products from and to MNE throughout the world. This lack of information leads us to take, as a first estimation, the simplest procedure, that is, to allocate emissions to MNE depending on

the presence of MNE in country c in every sector of every country (recorded in percentage terms in vector $\mathbf{m}_o^c$, see Supplementary Data 8–11).

CF and PF measures are the starting point for the calculation of the MNEs' carbon footprint, and according to those measures, we obtain two different measures depending on the allocation criteria of each (emissions embodied in exports to the country of consumption or to the country of production). On the one hand, from PF, we obtain the **MNE PF** of the MNE of country c allocated by the host country or, in other words, allocated to the sector and country where the production is occurring, as is shown in expression (1):

$$\mathbf{MNEPF} = \hat{\mathbf{e}}(\mathbf{I} - \mathbf{A})^{-1}\hat{\mathbf{m}}_o^c\mathbf{y} \hat{=} \mathbf{P}\hat{\mathbf{m}}_o^c\hat{\mathbf{y}} \qquad (1)$$

where $\mathbf{P} = \hat{\mathbf{e}}(\mathbf{I} - \mathbf{A})^{-1}$ is the emission multiplier matrix; $\hat{\mathbf{m}}_o^c$ is the diagonalized matrix of the percentages of every sector of production that originates from the outward MNE (indicated by the subscript o) of country c (indicated by the superscript c, being c = 1 … r) operating in every country throughout the world; and $\hat{\mathbf{y}}$ is the diagonal matrix of the final demand summed by rows. For instance, focusing on the U.S. empirically, the U.S. is country c; thus, MNE PF shows the carbon footprint of the US-MNE established in China, Australia or France. Expression (1) shows the total carbon embodied in the production of final goods and services supplied by the MNE corporations of a country's shareholders that produce and generate added value and employment beyond its borders. The lack of data regarding the detailed input structure of outward MNE implies that expression (1) is implicitly assuming that the production techniques of MNE are the same as the average sector in the host country. In other words, our calculations provide the MNE footprint in the special case in which MNE does not show any difference in production technique compared to the rest of firms within the sector[60] and[61] use the same assumption for other purposes, because they assume the same production structure of both non-exports and domestic production, without distinction of foreign-owned enterprises[38,62] distinguish this last category, but again they use the same assumption, the production structure of foreign invested enterprises in China is the same as the sector average (although they separate before processing exports). We overcome this assumption in scenarios 1 and 2, as explained in the following. In addition, the lack of data forced us to estimate m percentages as the share of MNE in the value added, wages, capital compensation and employment of the hosting sector in each country as proxies for their share in the sector emissions.

By contrast, the carbon footprint of the MNE allocated to the final consuming country (MNE CF) is obtained if we start from the CF as is shown in expression (2) as follows:

$$\mathbf{MNECF} = \hat{\mathbf{e}}(\mathbf{I} - \mathbf{A})^{-1}\hat{\mathbf{m}}_o^c\hat{\mathbf{y}}^{rr} + \hat{\mathbf{e}}(\mathbf{I} - \mathbf{A})^{-1}\hat{\mathbf{m}}_o^c\hat{\mathbf{y}}^{rs} = \mathbf{P}\hat{\mathbf{m}}_o^c\hat{\mathbf{y}}^d + \mathbf{P}\hat{\mathbf{m}}_o^c\hat{\mathbf{y}}^e \qquad (2)$$

where $\hat{\mathbf{y}}^d$ is the diagonal (in main diagonal) final demand domestically produced and consumed and $\hat{\mathbf{y}}^e$ is the diagonal (in off-diagonal positions) final demand exported (imported). Summing along the rows the **MNE CF** matrix, we have the same result as summing along the rows of **MNE PF** both for the sectors and countries, but the total for the countries and sectors are different when we sum down the columns because of the different allocation criterion.

**Accounting for different technical structures**. The procedure described implies that MNE and the national industries where the former operates share the same technical structure and the same share of imports and exports, implying also that the MNE supply does not differ from the average of the sector nor in type of product, quality or price, which constitutes a limitation of the analysis. Another limitation is that the model does not allow considering the different purposes of Foreign Direct Investment (such as access to new markets or resources, take advantage of lower labor costs or better suppliers[63]). Regarding the first limitation, the previous calculations provide the measure of the MNEs' carbon footprint in this counterfactual[64,65] if both industries, the domestic ones and MNE, share the same characteristics. Three strategies have been used to evaluate/reduce the importance of this assumption: (a) Some authors have evaluated how results change when a sector aggregation or disaggregation is held[38,59]; (b) Other researchers have developed hybrid LCA and Input-Output models[66] and obtained better results for hybrid models than with MRIO models; (c) others consider different value added content of exports regarding the size and ownership for Chinese firms[56] or different value added ratio and intensity of exports and imports[67].

We widened the analysis to include the possibility of a different technical structure for MNE with two alternative scenarios assuming that the host countries produce similarly to the home country (U.S.). In scenario 1 (S1), we replaced the input structure (columns of the technical coefficients matrix A), and in the scenario 2 (S2), we replaced both the input structure and the emission intensity (emissions coefficients of vector e). Both changes were applied only for the first production layer (initial and direct impact $(\mathbf{I} + \mathbf{A}^c)$) using structural path decomposition analysis[68]. The rest of the production chain (indirect impact, $\mathbf{L}^R = \mathbf{A}^2 + \mathbf{A}^3 + \mathbf{A}^4 + \mathbf{A}^5$) is the original one. This procedure reflects that the direct inputs used by the MNE and its direct emissions intensity are the same as they would be if the MNE was producing at country c, but the indirect inputs involved and their embodied emissions are those of the host country. This is very relevant in the case of electricity. A MNE could be as environmentally friendly as its headquarters in direct terms, but if the MNE is producing in China, for instance, the electricity the

MNE uses will be produced by the Chinese electricity sector, and it will be as polluting as that sector. As a result, the MNE' footprint by host and the consuming country under scenario 1 are provided by expressions (3) and (4), respectively:

$$\mathbf{MNEPF(S1)} = \hat{\mathbf{e}}(\mathbf{I} + \mathbf{A}^c)\hat{\mathbf{m}}_o^c\hat{\mathbf{y}} + \hat{\mathbf{e}}\mathbf{L}^R\hat{\mathbf{m}}_o^c\hat{\mathbf{y}} \qquad (3)$$

$$\mathbf{MNECF(S1)} = \hat{\mathbf{e}}(\mathbf{I} + \mathbf{A}^c)\hat{\mathbf{m}}_o^c\hat{\mathbf{y}}^{rr} + \hat{\mathbf{e}}\mathbf{L}^R\hat{\mathbf{m}}_o^c\hat{\mathbf{y}}^{rr} + \hat{\mathbf{e}}(\mathbf{I} + \mathbf{A}^c)\hat{\mathbf{m}}_o^c\hat{\mathbf{y}}^{rs} + \hat{\mathbf{e}}\mathbf{L}^R\hat{\mathbf{m}}_o^c\hat{\mathbf{y}}^{rs} \qquad (4)$$

Where $\mathbf{A}^c$ is the coefficient matrix with the inputs structure of the headquarters' country c, instead of the host country in sectors with MNE presence.

Under scenario 2 (S2), we replace both and the resulting MNE' footprint are given by expressions (5) and (6), respectively:

$$\mathbf{MNEPF(S2)} = \hat{\mathbf{e}}^c(\mathbf{I} + \mathbf{A}^c)\hat{\mathbf{m}}_o^c\hat{\mathbf{y}} + \hat{\mathbf{e}}\mathbf{L}^R\hat{\mathbf{m}}_o^c\hat{\mathbf{y}} \qquad (5)$$

$$\mathbf{MNECF(S2)} = \hat{\mathbf{e}}^c(\mathbf{I} + \mathbf{A}^c)\hat{\mathbf{m}}_o^c\hat{\mathbf{y}}^{rr} + \hat{\mathbf{e}}\mathbf{L}^R\hat{\mathbf{m}}_o^c\hat{\mathbf{y}}^{rr} + \hat{\mathbf{e}}^c(\mathbf{I} + \mathbf{A}^c)\hat{\mathbf{m}}_o^c\hat{\mathbf{y}}^{rs} + \hat{\mathbf{e}}\mathbf{L}^R\hat{\mathbf{m}}_o^c\hat{\mathbf{y}}^{rs} \qquad (6)$$

Where $\mathbf{e}^c$ is the vector of emissions coefficients assuming that host countries sectors of MNE pollute as headquarter's country.

**Materials**. We combined two main sources of data: the world input-output database (WIOD)[69] and the Bureau of Economic Analysis (BEA)[70] (see Supplementary Note 1). The first provides all relevant data for the construction of the basic and carbon-extended MRIO model, with 41 countries (see Supplementary Fig. 1) and regions and 35 sectors for 2009. The second provides information regarding US-MNE and affiliates operating outwards. There is a lack of reliable statistical information that provides detailed information from countries and sectors regarding the input structure of MNE. Focusing on the case of the U.S., as the BEA provides detailed information on U.S. multinationals from 16 sectors and all the countries for several magnitudes, we used value added, compensation of employees, capital compensation and employment of the majority-owned affiliates (affiliates that are owned 50% or more by foreign direct investors) operating abroad to estimate the shares of the MNE in each country and sector (see Supplementary Note 2 and Supplementary Data 8–11). Data about sectors and countries are suppressed to avoid the disclosure of data of individual companies and we take them as zero. As a result, we underestimated the US MNE' carbon impact in these particular countries and sectors where only one U.S. company is operating and where the foreign affiliates own 50% or less. Accounting for the majority-owned affiliates only implies considering 82% of all foreign affiliates' employment during 2009.

## Data availability

The authors declare that all the data (as well as the operational code) supporting the findings of this study are available within the paper and its supplementary information files, as follows:

The source data underlying all the Figures and tables in the Manuscript are provided as Supplementary Data files.

The code used in this text to perform the MRIO analysis is provided as Supplementary Data 16. However, the MATLAB files are available directly from the authors on request. The WIOD data (release 2013) that support the findings of this study is available in the website [http://www.wiod.org/release13].

The BEA data that support the findings of this study is available in the website [https://www.bea.gov/international/di1usdop].

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

## Acknowledgements

This work was supported by Ministerio de Economía y Competitividad (ECO2016–78938-R) and by the University of Castilla-La Mancha (UCLM).

## Author contributions

L.-A.L., M.-A.C., and J.Z. designed the study. J.Z. and G.A. performed the analysis and prepared the manuscript. All authors (L.-A.L., M.-A.C, J.Z., and G.A.) participated in the interpretation of data and in the writing of the manuscript.

## Additional information

**Competing interests:** The authors declare no competing interests.

