## [Peer Review File · Nature Communications]

Reviewers' comments:

Reviewer #1 (Remarks to the Author):

Referee Report

This paper studies the carbon footprint of multinational enterprises (MNEs) in the U.S. Specifically, the paper decomposes the carbon footprint along the MNEs' global value chains (GVCs) by applying a multiregional input-output model model to 2009 data. I have the following major concerns.

1. First and foremost, the study of carbon footprint along the global value chain using input-output methodology has received a lot of attention in the past decade. Conceptually, what is the main contribution of the study? How is the paper different from the existing studies in methodology?
2. Page 3, line 81-82, the paper claims that, "In the case of the U.S. economy, a reduction in 2007-2013 domestic CO₂ emissions has been mainly motivated by the influence of international trade, not only by new fracking techniques," based on reference [37]. However, this claim is not an accurate statement of the cited paper, which highlights that economic recession is the main reason behind emission reduction and change of fuel mix plays a minor role.
3. The study is based on data from year 2009, which is almost ten years ago. Why year 2009 is chosen for the study? Would the results still hold if the year is not 2009? It is well known that 2009 is the year right after the financial crisis, so would the policy implications drawn from 2009 be applicable in recent years? BEA provides more recent data. Why not use recent data? The significance of the results rest on the answer to these questions.
4. The use of BEA data and operationalization of the variables should be explained in more details. BEA offers various types of data for MNEs. As I understand, the data used are mainly from "Activities of U.S. Multinational Enterprises" section. This should be explained in details.

There are quite some minor concerns:

1. It is better to use abbreviation MNE rather than MN to denote multinational enterprises.

2. Line 149, "the US-MN' footprint is more similar to large polluting...". The sentence is confusing. How could the US MNEs' footprint be more similar to Japan than to South Korea? In the figure, it is closer to South Korea.

3. Line 354, "The MN' carbon footprint shifts...". The sentence is too long and a bit confusing.

4. Line 375, "carbon control Kagawa, Suh...", typo.

5. Line 433, "MN' carbon footprint...", typo.

6. Line 480, "and obtaining better results for hybrid models", should be "and obtained".

7. Line 507, "Data on sectors and countries suppressed...", typo.

There are quite some minor problems other than the above ones. The paper can benefit from a round of thorough proofreading.

Reviewer #2 (Remarks to the Author):

This article focuses on the carbon footprint of US multinationals. It has perfect logical derivation, in-depth analysis, and rich charts. Its idea is also good, and that could draw very meaningful conclusions if the method is appropriate. But there is still a certain gap between theoretical derivation and reality. The input-output table generally measures macro phenomena, but the carbon footprint of multinational companies seems to be a relatively microscopic thing. I don't think this article combines the two well. The key formula in this paper is equation (4) in SI, which just measures a kind of decomposition of the carbon footprint of the final product y , I think, but does not represent the carbon footprint of multinational enterprises. In fact, most multinational companies produce intermediate products. Therefore, I suggest that authors use other terms to replace multinational enterprises (MN) and carefully consider the concept of "producer carbon footprint". Except above big concern, there are still some minor advises following:

1, the title of the article does not reflect the core concept of the US-MN's carbon footprint. When I just saw this topic, I thought it is studying the overseas carbon footprint of the United States's final use.

2, Every figure should added labels to show country/region's names.

3, The classification of the regions should be clearly stated in the part of material. For example, what areas does ROW include?

4, In line 26 in SI, what the "characteristic matrix-element" means? Instead it by "Block-matrix element" is better, I think.

5, line 82 in SI, "while expression (3)," is (3) should be (4)? Please check that.

Reviewer #3 (Remarks to the Author):

1)The paper is solid work, with figures and comparisons made all the way through the paper, but for a reader its simply too much.It becomes to monoton, where nothing really stands out.

2)I like to see manuscript which starts with a short and precise abstract. This one, is a bit too long and it does not motivate a potential reader to read it.

3)You make a point of 1.7% of US-Carbon footprint comes from the import, but you do not say if that figure would have been higher or lower if all these goods had been produced in US. Neither do you not say anything about how much emissions is exported from US through products made in the US.

REVIEWERS.

Reviewer 1:

Referee Report

This paper studies the carbon footprint of multinational enterprises (MNEs) in the U.S. Specifically, the paper decomposes the carbon footprint along the MNEs' global value chains (GVCs) by applying a multiregional input-output model to 2009 data. I have the following major concerns.

1. First and foremost, the study of carbon footprint along the global value chain using input-output methodology has received a lot of attention in the past decade. Conceptually, what is the main contribution of the study? How is the paper different from the existing studies in methodology?

The main contribution of the study is to calculate for the first time the carbon footprint of the MNE affiliates that are operating outside the US all around the world, including the entire global supply chain of these companies. This calculation is essential in a context where NDC are not enough to achieve the target of 2 degrees (even less the 1.5) and non-state agents and companies are increasingly involved in pursuing actions to fight against climate change, but it is hard to assess the implementation and compliance of their pledges and the actual pollution reductions achieved (if any). Besides, the advantages of the inclusion of the supply chain of the MNE are threefold. First, as scope 3 is increasingly included in companies' pledges, our procedure allows including the scope 3 entirely avoiding problems of comparison when some activities in this scope are included and others do not. Second, by leveraging their supply chain power MNE can drive transnational corporate sustainability and carbon management improvements. And third, as most of the company's actions recorded are in developed countries, the presence of MNE affiliates in developing ones imply benefits there, also. In terms of methodology, our proposal also new although rely on common procedures in the input-output analysis for similar purposes (please see also an explanation on the support of the key formula give to reviewer #2).

We have modified some paragraphs of the introduction to clearly highlight the paper contribution and its relevance regarding existing literature on firm heterogeneity in input-output analysis, firm commitments on climate action, footprint calculations and the issue of supply chains and the inclusion of scope 3 and in terms of mitigation policy. As the changes we have made are in different paragraphs, please, see the specific modifications in the introduction section of the manuscript file.

2. Page 3, line 81-82, the paper claims that “In the case of the U.S. economy, a reduction in 2007-2013 domestic CO₂ emissions has been mainly motivated by the influence of international trade, not only by new fracking techniques,” based on reference [37]. However, this claim is not an accurate statement of the cited paper, which highlights that economic recession is the main reason behind emission reduction and change of fuel mix plays a minor role.

That is correct, thank you. We have rephrased the sentence accordingly.

3. The study is based on data from the year 2009, which is almost ten years ago. Why year 2009 is chosen for the study? Would the results still hold if the year is not 2009? It is well known that 2009 is the year right after the financial crisis, so would the policy implications drawn from 2009 be applicable in recent years? BEA provides more recent data. Why not use recent data? The significance of the results rest on the answer to these questions.

The reviewer is right about the period. However, we have included a justification for the chosen year for the analysis in the Supporting Information, in a new section called “**Materials processing**”. In this point we highlight the problem of the availability of MRIO data, global emission data and other studies that make similar decisions, such as Meng et al (2018) which analyze the burden of CO₂ emissions reductions of large Chinese companies based on data for the year 2010, justified by some reasons that can be extrapolated to our analysis such as the stability of the economic structure during the period and the stability of the energy use structure (especially in the period 2009 – 2014).

4. The use of BEA data and operationalization of the variables should be explained in more details. BEA offers various types of data for MNEs. As I understand, the data used are mainly from “Activities of U.S. Multinational Enterprises” section. This should be explained in details.

Thank you for the suggestion, we have extended the explanation about the data processing regarding the Multinationals data provided by BEA. There is a new section in the Supporting Information entitled “**Materials processing**”.

There are quite some minor concerns:

We thank the reviewer for his detailed corrections. All the typos and minor concerns of the reviewer have solved.

1. It is better to use abbreviation MNE rather than MN to denote multinational enterprises. Changed in the main text and the supporting information file.

2. Line 149, “the US-MN’ footprint is more similar to large polluting...”. The sentence is confusing. How could the US MNEs’ footprint be more similar to Japan than to South Korea? In the figure, it is closer to South Korea. We thank the reviewer for the note. We have changed the comment in the manuscript; the reviewer is right and the interpretation of the results in this part was inaccurate.

3. Line 354, “The MN’ carbon footprint shifts...”. The sentence is too long and a bit confusing.
We have simplified the sentence in the main text.

4. Line 375, “carbon control Kagawa, Suh...”, typo.

We have corrected the typo.

5. Line 433, “MN’ carbon footprint...”, typo.

We have corrected the typo.

6. Line 480, “and obtaining better results for hybrid models”, should be “and obtained”.

We have corrected the typo.

7. Line 507, “Data on sectors and countries suppressed...”, typo.

We have corrected the typo.

There are quite some minor problems other than the above ones. The paper can benefit from a round of thorough proofreading.

We are very sorry for all the typos, mainly considering that the paper has been sent to be revised by a service with native English speakers (American Journal Experts).

Reviewer 2:

This article focuses on the carbon footprint of US multinationals. It has perfect logical derivation, in-depth analysis, and rich charts. Its idea is also good, and that could draw very meaningful conclusions if the method is appropriate. But there is still a certain gap between **theoretical derivation and reality**. The input-output table generally measures macro phenomena, but the carbon footprint of multinational companies seems to be a relatively microscopic thing. I don't think this article combines the two well. The key formula in this paper is equation (4) in SI, which just measures a kind of decomposition of the carbon footprint of the final product y , I think, but **does not represent the carbon footprint of multinational enterprises. In fact, most multinational companies produce intermediate products**. Therefore, I suggest that authors **use other terms to replace multinational enterprises (MN)** and carefully consider the concept of “producer carbon footprint”.

First of all, thank you for your kind words about the quality of the derivation, analysis, charts, and conclusions of our work. Besides, we appreciate the effort of the reviewer in detecting potential flaws. We appreciate also the suggestion of the reviewer about the terminology used in the paper. Specifically, we have decided a change accordingly and use “carbon footprint of US MNE affiliates”. However, we have decided to keep the footprint denomination, because in the input-output framework this concept is implicitly linked to the total impact, directly and indirectly, triggered by the final demand. In addition, the footprint measure in multiregional input-output models avoids double counting, of emissions in this case ¹.

Second, we agree with the reviewer on his concern about the MNE that produce **intermediate products**. Our calculations are not based on a decomposition (or kind of) of the final demand (y), but on a decomposition on the carbon footprint of each sector (that is, total emissions directly and indirectly required to produce the sector final product) and we assume that the share of this footprint that corresponds to the MNE affiliate depends on the value added generated by the MNE in the sector in the country. Always in footprint calculations, the emissions (or all other impact or environmental pressure considered) embodied in intermediate products are allocated to the final product that requires them (as part of its production process). Obviously, this implies that the responsibility on emissions of the sectors producing intermediate products is “diluted” to the sectors that use those intermediate products. In other words, the responsibility of emissions is transferred from the sectors (and firms) producing intermediate products to those producing final products. As a result, sectors such as Construction shows higher carbon footprint (total emissions generated in the economy in the production of buildings, for instance) than the emissions the sector shows as a producer (emissions of its own production process). Something similar happens with footprints at country level and production-based inventories or consumption-base inventories or footprint: following this last one, emissions embodied in intermediate products and imports are

allocated, not to the sector and country that produces them (production inventories), but to the sector and country that consume the final product to have consumption-based inventories or footprint. The result is that countries specialized in highly polluting intermediate products for exports show lower carbon footprints (total emissions embodied in their economy final demand) than their territorial or production-based emissions. We will all agree that these facts do not question the validity or usefulness of the footprints calculations (so far).

Talking with other experts in input-output in a seminar at the NTNU, the issue of sectors and firms producing intermediates came up and we thought that it would be interesting to test an individual assessment of their emissions responsibility using, for instance, the hypothetical extraction method (used very successfully recently in the assessment of the impact of BREXIT in the UK and EU economies, ²), but this is far beyond the scope of the paper.

To estimate the size of this concern regarding US-MNE affiliates that produce intermediate inputs, we have calculated the differences between the producer responsibility (production-based inventories) and the carbon footprint of US-MNE in every sector and every country. This difference provides information on the extent of the distribution of emissions from intermediate inputs producers MNE, embodied in its inputs, to other sectors (MNE or not) that uses them directly or indirectly for production. The results show a quite wide range of variation. US-MNEs affiliates operating in the sector of primary and fabricated metals show a carbon footprint that varies from 2% of its production-based emissions in Indonesia to 203% in Portugal. Mining US-MNE carbon footprint over its production-based inventory varies from 1% in Brazil, Canada or Russia to the 523% of Ireland or the 63% in Austria and Other industries (that is the aggregation of electricity and construction, unfortunately, two sectors very different regarding carbon footprint and production-based accounting) between 66% in The Netherlands to 130% in Austria.

Third, regarding **the key formula** of the paper, as our paper does, there are several examples in the literature that deal with firm heterogeneity inside the input-output framework, that is, that try to split the sector average considering different types of firms. Their goals are diverse, but they focus on the Chinese economy and are usually related to the phenomenon of global value chains and international trade, from the assessment of the carbon emissions to the value-added embodied in international trade (^{3 4 5 6 7 8 9}). These examples mostly imply fully splitting a Chinese (or the world in the last case) input-output table according to the type of production (processing exports or not), the ownership of firms or the firm size, by using firm-level data from outside the National Accounts, in some cases.

In all those cases, the use of a proportionality distribution criterion is the initial strategy to estimate different input-output columns (that show production technique and inputs used by every sector) and as a result, the technique and inputs of the different types of firms considered. The differences among the papers are related to which measure they use for the proportionality assumption and, when they use external information to the input-output framework, the method to reconcile results with the standard input-output identities (RAS or similar in some cases or quadratic programming in others). In each case, the share criterion depends on the detail provided by the firm data source (usually from other sources and outside the national accounts system and methodology) about production, value added and intermediate consumption. If there is available firm information, specifically, on production and value added, they calculate the intermediate consumption as a difference (notice that in this case, intermediate consumption is obtained as an aggregated amount). Then, the volume of intermediate consumption is distributed among different inputs following the same share of the average.⁹, that indicate that OECD is building an input-output table of multinationals for several countries, split between domestic inputs and imported ones using the share on the total sector production (from input-output tables) of production by domestic-owned firms and by foreign-owned firms, obtained from firm data sources.

In our case, BEA does not provide data on total production or intermediate consumption of MNE, so we cannot calculate intermediate consumption by difference or instead of that use total production to distribute the sector intermediate consumption. Thus, we have decided to allocate to MNE the share of the sector carbon footprint using the share of MNE in the value added of the sector. Besides, to evaluate how the choice of the variable to split up the carbon footprint of the sector affect the results, we make the estimations using 4 different variables, monetary (value added, wages and capital compensation) and physical (employment). On the one hand, since the carbon footprint concept implies that emissions are linked to the emissions embodied in the inputs required directly and indirectly for production, our procedure implicitly assumes that the share of MNE on the specific sector value added is a good proxy of the share of MNE in the total (direct and indirect) intermediate consumption of the sector. It also implies that that MNEs show the same share of intermediate consumption over value added that the specific sector average. On the other hand, our calculations provide the MNE footprint in the special case in which MNE does not show any difference in production technique compared to the rest of firms within the sector and show the same technique than the average.³ and ⁴ use the same assumption for other purposes, because they assume the same production structure of both non-processing exports and domestic

production, without distinction of foreign-owned enterprises (as confirmed also by ¹⁰). ⁵ distinguish this last category, but again they use the same assumption, the production structure of foreign-invested enterprises in China is the same as the sector average (although they separate before processing exports). We go a step further because through two different scenarios we change this assumption and assess its impact on the MNE carbon footprint. Another difference of our proposal compared to previous cited literature, is that they, once split the intermediate matrices, need to balance them in order to fulfill the input-output basic identities, as commented before, and as a result of this balancing process the techniques of each type of firm end to be different to each other. Our proposal does not need this balance, because none of the identities are broken since we maintain the same production technique.

In a different research line, the proposal of ¹¹ includes also a proportionality assumption using a diagonal matrix of shares (intermediate inputs of each industry supplied by domestic production) to estimate the displacement of emissions by imports including intermediate imports (this is the equation (8) in ¹¹ of emissions displaced by imports) because he does not have the intermediate imports matrices provided nowadays by multiregional databases. This implies assuming that the fraction of any input that is imported is the same, regardless of which industry uses it. This was a quite common assumption when intermediate imported inputs matrices were not available.

Sorry for the extensive “answer” to this issue, but as you said below is a big concern and it deserves a big answer (and solid foundation).

Except above big concern, there are still some minor advises following:
1, the title of the article does not reflect the core concept of the US-MN’s carbon footprint. When I just saw this topic, I thought it is studying the overseas carbon footprint of the United States’s final use.

The title has been modified, now we use the term “affiliates” in order to guide the reader in a better understanding of the manuscript’s content.

2, Every figure should added labels to show country/region’s names.

We have included a footnote to each figure with the reference to the part of the Supporting Information where you can find the different regions/countries in this study (please, see also the answer to the following comment).

3, The classification of the regions should be clearly stated in the part of material. For example, what areas does ROW include?

We have included in the Supporting Information a new section explaining the databases used and the treatment of the data, where we explain the countries and regions considered in the study. In the mentioned section, we have included a map where the areas considered within the RoW are colored in gray and the way in which they have been calculated and the references where they are explained in detail are specified.

4, In line 26 in SI, what the “characteristic matrix-element” means? Instead it by “Block-matrix element” is better, I think.

We agree and we have changed it.

5, line 82 in SI, “while expression (3),” is (3) should be (4)? Please check that.

The reviewer is right, it is a typo and we have solved it. Thank you.

Reviewer 3:

1) The paper is solid work, with figures and comparisons made all the way through the paper, but for a reader its simply too much. It becomes too monoton, where nothing really stands out.

We agree with the reviewer that conveying the results of researching to a wider audience is one of the important challenges of researchers, and a hard one. It is hard because we deal with complex problems and complex methods and because we are not very used to it, also. We try to do it using comparisons, as reviewer #3 points out. We have modified some paragraphs in the introduction to make it more appealing and highlighted the novelties and contributions of the paper.

2) I like to see manuscript which starts with a short and precise abstract. This one, is a bit too long and it does not motivate a potential reader to read it.

We have modified the abstract following the suggestions of the reviewer.

3) You make a point of 1.7% of US-Carbon footprint comes from the import, but you do not say if that figure would have been higher or lower if all these goods had been produced in the US. Neither do you not say anything about how much emissions is exported from the US through products made in the US.

We think there is a little misunderstanding with this figure in particular. This 1.7% refers to the carbon footprint of the US Multinationals affiliates enterprises beyond their borders. It does not refer to imports and, therefore, in our opinion, it would not make sense to compare with exports from the US. In addition, this would be a totally different approach to the work that we intend to do since then we would be talking about affiliates multinational companies located in the US to meet the global final demand. Certainly, it is an interesting work proposal for further research.

However, we have found very interesting the proposal of the reviewer on the approach of a counterfactual and we have calculated emissions that would be generated if the production of the MNE abroad was produced in the US and introduced a short comment of the results into the manuscript (specifically, in the scenarios and indicators section). Our results show that, if instead of occurring in other countries those goods and services were produced in the US, the emissions would be slightly lower, representing 1.58% of the worlds' global emissions in 2009, instead of 1.7% that represents the carbon footprint of US-MNE beyond their borders.

References cited in this document

1. Lenzen M, Lundie S. Constructing enterprise input-output tables - a case study of New Zealand dairy products. *Journal of Economic Structures* **1**, 6 (2012).
2. Chen W, Los B, McCann P, Ortega-Argilés R, Thissen M, van Oort F. The continental divide? Economic exposure to Brexit in regions and countries on both sides of The Channel. *Papers in Regional Science* **97**, 25-54 (2018).
3. Koopman R, Wang Z, Wei S-J. How Much of Chinese Exports is Really Made In China? Assessing Domestic Value-Added When Processing Trade is Pervasive. *National Bureau of Economic Research Working Paper Series No. 14109*, (2008).
4. Koopman R, Wang Z, Wei S-J. Estimating domestic content in exports when processing trade is pervasive. *Journal of Development Economics* **99**, 178-189 (2012).
5. Dietzenbacher E, Pei J, Yang C. Trade, production fragmentation, and China's carbon dioxide emissions. *Journal of Environmental Economics and Management* **64**, 88-101 (2012).
6. Su B, Ang BW, Low M. Input-output analysis of CO2 emissions embodied in trade and the driving forces: Processing and normal exports. *Ecological Economics* **88**, 119-125 (2013).
7. Tang H, Wang F, Wang Z. The Domestic Segment of Global Supply Chains in China under State Capitalism. *Policy Research Working Paper, World Bank, Washington, DC 6960*, (2014).
8. Meng B, *et al.* More than half of China's CO2 emissions are from micro, small and medium-sized enterprises. *Applied Energy* **230**, 712-725 (2018).
9. Cadestin C, Backer KD, Desnoyers-James I, Miroudot S, Rigo D, Ye M. Multinational enterprises and global value chains: the OECD analytical AMNE database. (2018).
10. Xia Y, Fan Y, Yang C. Assessing the impact of foreign content in China's exports on the carbon outsourcing hypothesis. *Applied Energy* **150**, 296-307 (2015).
11. Levinson A. Technology, International Trade, and Pollution from US Manufacturing. *American Economic Review* **99**, 2177-2192 (2009).

Reviewers' comments:

Reviewer #1 (Remarks to the Author):

I thank the authors for their efforts in revising the paper. All my concerns have been addressed. I am recommending acceptance.

Reviewer #2 (Remarks to the Author):

The paper has been improved with the comments from the reviewers. The author replaced some concepts and made the article logically clear. In addition, the author has also modified the original method, and the method is clear and easy to understand after modification.

This round of comments, I am most concerned about the data source of the percentages of every sector production that comes from the outward MNE, says Mo. The author pointed out that they used four indexes of overseas activities from BEA, but steps how to obtain Mo need to further explain clearly, is it weighted average or simply summed up? Is it divided by the gross value-added in the input-output table to obtain the percentages, or the percentages just also come from BEA? These need to be clearly explained in the text. And I also recommend that the authors show the final Mo data of US in the SI.

Reviewer #3 (Remarks to the Author):

You have improved and modified the manuscript to satisfy the comments from all the reviewers.

REVIEWERS.

Reviewer 1:

I thank the authors for their efforts in revising the paper. All my concerns have been addressed. I am recommending acceptance.

Thank you very much.

Reviewer 2:

The paper has been improved with the comments from the reviewers. The author replaced some concepts and made the article logically clear. In addition, the author has also modified the original method, and the method is clear and easy to understand after modification.

Thank you very much.

This round of comments, I am most concerned about the data source of the percentages of every sector production that comes from the outward MNE, says Mo. The author pointed out that they used four indexes of overseas activities from BEA, but steps how to obtain Mo need to further explain clearly, is it weighted average or simply summed up? Is it divided by the gross value-added in the input-output table to obtain the percentages, or the percentages just also come from BEA? These need to be clearly explained in the text. And I also recommend that the authors show the final Mo data of US in the SI.

Thank you for the suggestion. Within the materials section in the Supporting Information, a new subsection, entitled "US-MNE value added shares", has been included. This subsection explains further the steps in the estimation of the m_o^c vectors and shows the final figures used in the calculations through four new different tables.

As we explain in the text, and trying to clarify the reviewer's concerns, to estimate the m_o^c vector we combine two main databases, BEA's AMNE database and WIOD. This combination is similar to those followed by 1 (that combine OECD AMNE and WIOD) and by 2 (that combine BEA's AMNE and U.S. SUT), although with different final goals and procedures. The combination of the two sources can provide some biases in the results at sector level (not in total country figures) related to the different unit considered, establishment in WIOD and enterprise in AMNE. The differences are reduced to the presence of several affiliates with activity in different sectors because the BEA explicitly identifies the primary industry of the controlling unit, as well as the primary industry of its affiliates, according to the type of products with the highest share in sales³. The calculations are as follows. First, we extract the US-MNE activity (value added, for instance) from BEA for each sector in each country (section 'Activities of U.S. Multinational Enterprises (MNEs)'). Second, we divide it by the total value added (or the appropriate indicator) of each sector in each country taken from WIOD to obtain the share of value added generated by the US-MNE affiliate activity. Although the main estimations of this paper are based on the value added (gross product) generated abroad by US-MNE affiliates, three other alternative indicators of US-MNE activity abroad have been taken into account: Compensation of Employees, Capital Compensation and Employment, following analogous calculations. Finally, four different m_o^c are estimated, one for each indicator of US-MNE activity. The m_o^c vectors are shown in Tables 1-SI, 2-SI, 3-SI and 4-SI in the supporting information and also are shown in the Excel Source file.

Reviewer 3:

You have improved and modified the manuscript to satisfy the comments from all the reviewers.

Thank you very much.

References:

1. Cadestin C, Backer KD, Desnoyers-James I, Miroudot S, Rigo D, Ye M. Multinational enterprises and global value chains: the OECD analytical AMNE database. (2018).
2. Fetzer JJ, Highfill T, Hossiso KW, Howells TF, III, Strassner EH, Young JA. Accounting for Firm Heterogeneity within U.S. Industries: Extended Supply-Use Tables and Trade in Value Added using Enterprise and Establishment Level Data. *National Bureau of Economic Research Working Paper Series No. 25249*, (2018).
3. BEA. U.S. International Economic Accounts: Concepts and Methods. (ed^(eds). Bureau of Economic Analysis, U.S. Department of Commerce. (2014).

REVIEWERS' COMMENTS:

Reviewer #2 (Remarks to the Author):

The article has been revised in accordance with my last comments, I have no other comments this round.